



# Isoscape of amount-weighted annual mean precipitation tritium ([3]H) activity from 1976 to 2017 for the Adriatic–Pannonian region – AP[3]H_v1 database

**Zoltán Kern**[1], **Dániel Erdélyi**[1,2], **Polona Vreča**[3], **Ines Krajcar Bronić**[4], **István Fórizs**[1], **Tjaša Kanduč**[3], **Marko Štrok**[3], **László Palcsu**[5], **Miklós Süveges**[6], **György Czuppon**[1,5], **Balázs Kohán**[7], and **István Gábor Hatvani**[1]

[1]Institute for Geological and Geochemical Research, Research Centre for Astronomy and Earth Sciences, MTA Centre for Excellence, Budaörsi út 45, 1112 Budapest, Hungary
[2]Centre for Environmental Sciences, Eötvös Loránd University, Pázmány Péter stny. 1/A, 1117 Budapest, Hungary
[3]Department of Environmental Sciences, Jožef Stefan Institute, Ljubljana, Slovenia
[4]Department of Experimental Physics, Ruđer Bošković Institute, Bijenička 54, 10000 Zagreb, Croatia
[5]Isotope Climatology and Environmental Research Centre (ICER), Institute for Nuclear Research, Bem tér 18/c, Debrecen, Hungary
[6]HYDROSYS Labor Ltd., Botond utca 72., 1038 Budapest, Hungary
[7]Dept. of Environmental and Landscape Geography, Eötvös University, Pázmány stny 1/C, 1117 Budapest, Hungary

**Correspondence:** István Gábor Hatvani (hatvaniig@gmail.com)

**Abstract.** Tritium ([3]H) as a constituent of the water molecule is an important natural tracer in hydrological sciences. The anthropogenic tritium introduced into the atmosphere unintentionally became an excellent tracer of processes on a time scale of up to 100 years. A prerequisite for tritium applications is to know the distribution of tritium activity in precipitation. Here we present a database of isoscapes derived from 41 stations for amount-weighted annual mean tritium activity in precipitation for the period 1976 to 2017 on spatially continuous interpolated $1\,\mathrm{km} \times 1\,\mathrm{km}$ grids for the Adriatic–Pannonian region (called the AP[3]H_v1 database), with a special focus on post-2010 years, which are not represented by existing global models. Five stations were used for out-of-sample evaluation of the model performance, independently confirming its capability of reproducing the spatiotemporal tritium variability in the region. The AP[3]H database is capable of providing reliable spatiotemporal input for hydrogeological application at any place within Slovenia, Hungary, and their surroundings. Results also show a decrease in the average spatial representativity of the stations regarding tritium activity in precipitation from $\sim 440\,\mathrm{km}$ in 1970s, when bomb tritium still prevailed in precipitation, to $\sim 235\,\mathrm{km}$ in the 2010s. The post-2010 isoscapes can serve as benchmarks for background tritium activity for the region, helping to determine potential future local increases in technogenic tritium from these backgrounds. The gridded tritium isoscape is available in NetCDF-4 at https://doi.org/10.1594/PANGAEA.896938 (Kern et al., 2019).

## 1   Introduction

Tritium ($^3$H) is a radioactive isotope of hydrogen (Alvarez and Cornog, 1939) with a half-life of 12.32 years (4500±8 d; Lucas and Unterweger, 2000). Natural tritium is formed mainly by spallation reactions of protons and neutrons of primary and secondary cosmic radiation with atmospheric nuclei, mainly by the interaction of fast neutrons with atmospheric nitrogen (Lal and Peters, 1967). Tritium emission by thermonuclear tests between the 1950s and 1980 enormously exceeded the natural production (Araguas-Araguas et al., 1996; Palcsu et al., 2018). Since that time, tritium emission to the atmosphere from anthropogenic sources (e.g., nuclear industry, medical applications, luminizing industry) has corresponded to ∼ 10 % of the natural production and influences $^3$H content in precipitation mainly at local to regional scales (Araguas-Araguas et al., 1996). Starting from the 1980s, technogenic tritium has become the prevailing anthropogenic atmospheric tritium input signal over bomb tritium in Central Europe (Hebert, 1990).

Tritium is introduced into the hydrological cycle following oxidation to tritiated water ($^3$H$^1$HO). Tritium is an excellent tracer for determining time scales for the mixing and flow of waters and is ideal for studying processes that occur on a time scale of fewer CEI than 100 years (Kendall and McDonnell, 2012). It proved to be a powerful tool in various applications in hydrological research (Jasechko, 2019), such as estimating mean residence time for surface water and groundwater (Michel, 1992; Stewart and Morgenstern, 2016; Zuber et al., 2001), dating cave drip waters (Kluge et al., 2010), understanding water circulation and mixing in geothermal (Ansari et al., 2017; Chatterjee et al., 2019) or permafrost settings (Gibson et al., 2016), and many other fields (Eyrolle et al., 2018).

A prerequisite for such applications is either a measured or modeled reference of precipitation tritium activity (Stewart and Morgenstern, 2016). Long-term measurements for precipitation tritium activity are rare worldwide, and even the longest time series are usually intermitted by gaps. In the absence of on-site measurements, either remote monitoring data have to be used as references (Huang and Pang, 2010; Thatcher et al., 1961), or estimations are required. There are several methods to reconstruct precipitation tritium time series for geographical locations (Li et al., 2019). The prediction of the first global model for tritium distribution in precipitation from 1960 to 1986 (Doney et al., 1992) was improved and provided a higher-accuracy estimate for precipitation $^3$H variations (Zhang et al., 2011) extending up to 2005 called the "modified global model of tritium in precipitation" (MGMTP). Unfortunately, the key parameters of the MGMTP are only available as isoline maps (Zhang et al., 2011), from which the model's coefficients can be extracted with high uncertainty in a manual way, which leads them to be ambiguous. In addition, the quality of the estimated precipitation tritium activity values by the MGMTP become

quite poor after 1990 (Zhang et al., 2011); for instance, in the studied region it produced uninterpretable, negative values (Sect. 4). The most recent global model for precipitation tritium activity covering the period 1955–2010 (Jasechko and Taylor, 2015) used inverse distance weighting for interpolation, and its output is available in gridded format. However, this is based only on precipitation $^3$H activity concentration records of the stations of the Global Network of Isotopes in Precipitation (Rozanski et al., 1991), and it does not represent the most recent decade.

Although global models are available, due to the differences in tritium activity around the globe it is beneficial to define local precipitation $^3$H input curves (Stewart and Morgenstern, 2016). In the northern part of the Balkan region, for instance, it was shown that $^3$H content in precipitation deviated from the Vienna record considerably after 1980 (Miljević et al., 1992). This record is popularly used as a remote reference station in hydrological modeling and calculations in the Adriatic–Pannonian region. The quality of such curves is vital for the reliability of a hydrological model's output when employed as an input signal or data in hydrological modeling and calculations (Koeniger et al., 2008; Miljević et al., 1992). Indeed, it has recently been found that the (in)accuracy of the precipitation tritium time series used is the key uncertainty factor for groundwater recharge estimations (Li et al., 2019).

Measurements of precipitation tritium activity in the Adriatic–Pannonian region began in Vienna Hohe Warte in 1961, which is the longest continuously operating station in Central Europe and in the world (IAEA, 2019). Additional stations started operation in the past ∼ 50 years with frequent interruption in data collection (Araguas-Araguas et al., 1996; Krajcar Bronić et al., 1998; Rozanski et al., 1991; Vreča et al., 2008). The demand for long-term precipitation $^3$H reference time series in various hydrological and hydrogeological applications across the Adriatic–Pannonian region called forth the use of remote stations (e.g Gessert et al., 2019; Kanduč et al., 2014, 2012) and/or motivated the derivation of case-specific "composite" tritium reference curves (e.g., Kern et al., 2009; Krajcar Bronić et al., 1992; Ozyurt et al., 2014; Szucs et al., 2015). Derivation of these ad hoc "composite" $^3$H reference curves usually applied different imputation methods to the "gappy" time series and/or employed different interpolation techniques. Differences in the absolute values due to methodological differences might seem marginal if the peak concentration of the mid-1960s is used as a time marker; however if data are used as input in tritium mass balance models in the postbomb period, differences become highly important (Li et al., 2019).

The aim of this study is to create a spatially continuous gridded database for tritium (isoscape) in precipitation across the Adriatic–Pannonian realm for the decades around the turn of the 21st century with a special focus on post-2010, which is not covered by the existing global models.

## 2  Materials and methods

### 2.1  Used $^3$H and precipitation data

An initial dataset was collected with 8450 monthly precipitation tritium activity values from 46 stations covering the period from January 1961 to December 2017. Tritium content is expressed in tritium units (TU), where 1 TU = 0.118 Bq L$^{-1}$. This dataset was obtained from the Global Network of Isotopes in Precipitation (GNIP; IAEA, 2019), the Austrian Network of Isotopes in Precipitation (Umweltbundesamt, 2019), the Slovenian Network of Isotopes in Precipitation (SLONIP; SLONIP, 2020; Vreča and Malenšek, 2016), and published data (Fórizs et al., 2020; Krajcar Bronić et al., 2020; Mandić et al., 2008; Palcsu et al., 2018). Until 1973 tritium activity data were only available from Austria. Monitoring of isotopes in precipitation on a larger scale in the region began in the mid-1970s in Belgrade (RS), Zagreb (HR), and Budapest (HU) as well. Following the initiation of these measurements the network becomes suitable – specifically from 1976 – for the spatiotemporal analysis of the large-scale variability in precipitation tritium activity in the region. To maximize the spatiotemporal density of the dataset, not only the Adriatic–Pannonian region but also the bordering areas were included in the analyses. The availability of $^3$H data varied in the investigated period. The relative abundance of data increased in the early 1980s, early 2000s, and from around 2010 onward (Fig. 1a). Between 2003 and 2005, the number of stations dropped (< 9; Fig. 1a) due to a halt in data collection at the Austrian stations. This was the lowest number of active stations in the investigated period. For the purpose of further calculations, the geographical coordinates of the stations were converted from latitude and longitude (EPSG: 4326; WGS84 projection) to the metric coordinate system (EPSG: 3857; WGS84/pseudo-Mercator projection) since interpolation (variography; see Sect. 2.3) has to be done on a metric scale. To be able to derive amount-weighted annual tritium activity averages, monthly precipitation amounts were used from the Global Precipitation Climatology Centre (GPCC; 1.0° × 1.0°) Full Data Monthly Product Version 2018 (Schneider et al., 2018).

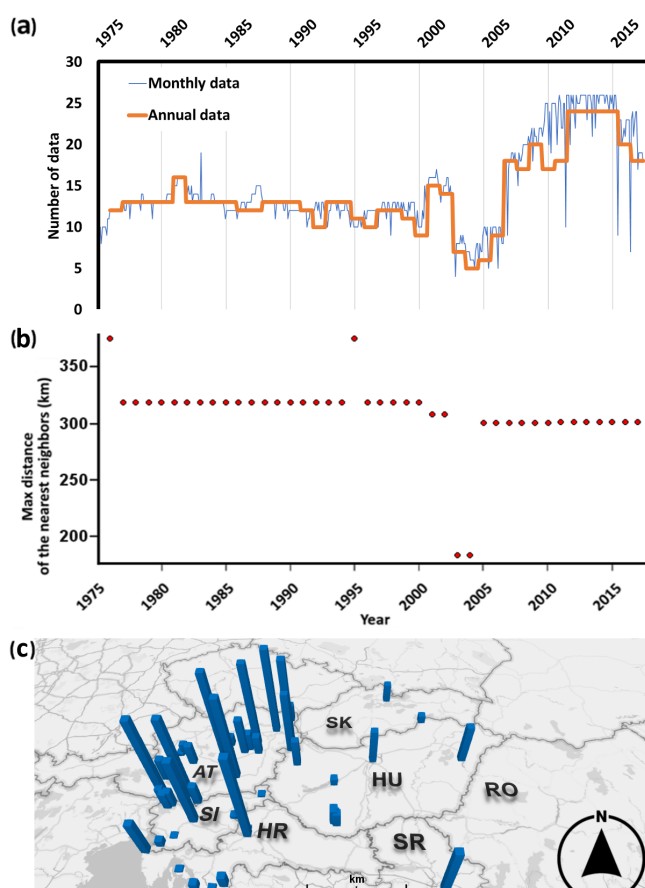

**Figure 1.** Temporal and spatial characteristics of the dataset. Number of data from precipitation stations producing measurements of $^3$H (1975–2017). **(a)** The thick orange line represents the number of stations applicable for computing amount-weighted annual precipitation averages later used in the interpolation (1976–2017). **(b)** The largest distance between the neighboring active stations of the studied $^3$H network in each year for 1976–2017. **(c)** Spatial distribution of the monitoring sites, where the height of the blue columns is proportional to the number of monthly data available between 1976 and 2017 at a given station; max = 479 data at Podersdorf, Austria. The country codes follow ISO-3166-1 ALPHA-2. The base map was taken from Bing maps, HERE Technologies 2019.

### 2.2  Data preprocessing

A sequential univariate outlier detection procedure (Ben-Gal, 2005) was applied to the data to find possible outlying values that deviate to a high extent from the other observations (Barnett and Lewis, 1974; Hawkins, 1980). During the procedure, the time series of the stations were compared pairwise for each year. The approach is similar to the relative homogeneity test applied to meteorological data, in which, for example, a candidate station's time series is compared to its neighboring stations' (e.g., Alexandersson, 1986; Lindau and Venema, 2019; Sugahara et al., 2012).

To avoid comparing a station with all the others from the network, including distant ones recording different environmental conditions (e.g., Alpine region vs. Great Hungarian Plain), the comparison was done only within a given search radius. The network was screened for each station's distance to its nearest neighbor for each year. Then out of all the years, the most frequently occurring largest and nearest neighbor (∼ 320 km) was chosen (Fig. 1b) to serve as the search radius for the sequential univariate outlier detection. There were only two years when a station – specifically Belgrade – did not have a pair to compare it with due to its relatively isolated location from the others in the network (Fig. 1c).

Pairwise differences of [3]H data in monthly steps were calculated for each station with its neighbors within the $\sim$ 320 km search radius. These pairwise differences were then averaged per month, and the values belonging to the same calendar year were handled together. Due to the decrease in atmospheric concentration of tritium (Palcsu et al., 2018; Rozanski et al., 1991), the difference values were not comparable between the years, so the outliers were identified annually. The monthly average difference values were annually standardized.

It was found that the standardized mean differences were mostly within the $\pm 1$ interval (89 %), suggesting a usually small difference between neighboring records. In rare occasions ($n = 11$ occurrences; 0.15 %) the difference value was outside the $\pm 7$ interval. These deviations were considered as a threshold, determining the set of possibly erroneous data (outlier; Ben-Gal, 2005), which were investigated one by one, if possible, by consulting the data providers. Only two monthly values were left out from further calculations, both much higher than the measured ones compared to their surroundings.

Annual amount-weighted means were only calculated if at least 85 % of the fallen precipitation was analyzed for [3]H; these years are referred to as "complete years" herein. If more than 15 % of the fallen precipitation was not analyzed for [3]H, the year in question is referred to as an "incomplete year". This required completeness is a stricter criterion than the GNIP protocol (70 %; IAEA, 1992). These amount-weighted annual averages served as the input values for deriving the isoscapes with variography. An additional check was performed on the amount-weighted annual means using h-scattergrams (Bohling, 2005), which did not find any outliers that have been introduced by the weighting procedure, confirming that the dataset satisfies certain prerequisites of kriging.

A robust hemispheric-scale pattern is a poleward-increasing trend of precipitation [3]H (Rozanski et al., 1991). Regression analysis between geographical latitude (using the metric coordinates in EPSG: 3857) and amount-weighted annual precipitation [3]H activity concentration mostly yielded insignificant linear relationships or contradictory relationships to what was expected (i.e., poleward-decreasing values in the year 1987, for example). The limited latitudinal extent of the study area (5°) might explain the failure to detect the expected relationship. However, due to the lack of a clear spatial trend, statistical trend removal was not conducted on the amount-weighted annual mean [3]H activity. Instead they were used for regional isoscape modeling.

## 2.3 Derivation of amount-weighted annual mean precipitation tritium activity isoscapes

Semivariograms (Webster and Oliver, 2008) were used as the weighting function in kriging (Cressie, 1990) to explore the spatial variance of amount-weighted annual mean precipitation [3]H activity for the stations of the Adriatic–Pannonian region. The empirical semivariogram may be calculated using the Matheron algorithm (Matheron, 1965), where $\gamma(h)$ is the semivariogram, and $Z(x)$ and $Z(x + h)$ are the values of a parameter sampled at a planar distance $|h|$ from each other:

$$\gamma(h) = \frac{1}{2N(h)} \sum_{i=1}^{N(h)} [Z(x_i) - Z(x_i + h)]^2. \tag{1}$$

$N(h)$ is the number of lag-$h$ differences, i.e., $n \times (n-1)/2$, and $n$ corresponds to the number of sampling locations at a distance $h$. The most important properties of the semivariogram are (1) the nugget, which quantifies the variance at the sampling location (including information regarding the error of the sampling); (2) the sill, that is the level at which the variogram stabilizes, which is the sum of the nugget ($c_0$) and the reduced sill ($c$); and (3) the range ($a$), which is the distance within which the samples have an influence on each other and beyond which they are uncorrelated (Chilès and Delfiner, 2012). If the semivariogram does not have a rising part and the points of the empirical semivariogram align parallel to the abscissa, a nugget-type variogram is obtained. In this case, the sampling frequency is insufficient to estimate the sampling range using variography (Hatvani et al., 2017).

For geostatistical modeling (e.g., kriging), theoretical semivariograms have to be used to approximate the empirical ones (Cressie, 1990). Gaussian semivariograms were obtained with a maximum lag distance of 400 km and 11 uniform bins (steps) in order to achieve the most balanced number of station pairs per bin in the analysis. The effective (or even called practical) range ($a_e$), which is the distance within which the samples have an influence on each other and beyond which they are uncorrelated (Chilès and Delfiner, 2012), was determined and used to evaluate the spatial representativity of the network. In the case of Gaussian semivariograms $a_e = \sqrt{3} \times a$ (Wackernagel, 2003). The reported ranges in the study area are planar distances in kilometers; conversion to geodetic distance in the region can be done using the approximation $d_{planar} \times 0.678 \approx d_{geodetic}$.

Semivariograms applicable for interpolation were obtained from years 1977, 1982, 1983, 2001, and 2010 as well as from 2012 to 2015, which, unsurprisingly, belonged to the periods with a relatively higher abundance of data (Fig. 1a). Common characteristics of the variograms from these years (herein referred to as the "reference years") were that (i) the number of stations behind the variograms from these years varied between 13 and 24, and (ii) the variograms had at least 11 pairs in the first three bins without any empty ones. This seemed to be a minimum requirement in the present database to derive theoretical variograms suitable for kriging.

The years with a reduced number of available stations (Fig. 1a) produced semivariograms not applicable for kriging (for a technical explanation see Appendix A) because the data were sporadically spread in space, and/or none of the stations provided continuous measurements in time. Both types of data gaps can be classified as missing at random

(MAR) (Little and Rubin, 2002). Because most modern data imputation methods start by assuming the missing data are MAR, imputation tools could have been applied in years with insufficient data density for proper interpolation. However, in every case, no method can provide an "automatic" solution to the problem of missing data, and any approach must be used with caution considering the context of the problem (Kenward and Carpenter, 2007); for instance, the accuracy of the imputed value will not be optimal, and the spatial correlation and intravariable relationships will be corrupted (Barnett and Deutsch, 2015).

Thus, in these so-called "intermediate" years, the semivariogram of the reference years with the most overlap with regard to its station distribution was used as the weight for kriging. To do so, for each intermediate year, the number of sites that are commonly active in its temporally neighboring reference year was investigated. The following requirements were also considered:

- The maximum number of sites active in a given intermediate year which are not active in the reference year can be three.

- If the difference in the number of active stations between an intermediate year and the "neighboring" two reference years is the same, then the semivariogram of the reference year with the greater number of active stations was used, rendering that variogram more robust, and/or the variogram from the reference year closest in time to the intermediate year was used.

Finally, 46 stations were considered for further evaluation, out of which 41 stations were used for tritium isoscape derivation. The amount-weighted annual tritium activity from these 41 stations were used to derive the gridded ($1\,\mathrm{km} \times 1\,\mathrm{km}$) annual precipitation $^3$H isoscapes across Slovenia and Hungary for 1976–2017, called the AP$^3$H_v1 database, using ordinary point kriging with the assigned variograms derived as discussed above. Note here that the grid resolution was chosen based on practical consideration; it is not intended to imply that there are such fine kilometer-scale differences but helps the users to delineate smaller outcrops of, for example, watershed areas more accurately.

For out-of-sample verification, two stations were withheld – Nick (HU; active: 1990–2004) and Zgornja Radovna (SI; active: 2010–2017) – to validate that the interpolated product is useful and, in a sense, justifies the application of kriging in the region. Two additional stations recorded precipitation $^3$H for multiple but exclusively incomplete years: Jósvafő (HU; active: 1988–1996) and Malinska (HR; active: 2000–2001). These stations were not applicable for variography yet were used to further test the performance of the geospatial model of precipitation $^3$H and the critical limit of the ratio of annual precipitation amount with missing $^3$H data. Lastly, an additional short record from Siófok (HU; active: 2013–2016) was also excluded from the spatial model to compare it with the interpolated data (see Sect. 4).

All computations were performed with Golden Software Surfer 15, ArcGIS 10, GS+ 10, and R (R Core Team, 2019) using the script in Supplement. For certain visualizations of the results, Gimp 2.8 and MS Excel 365 were used.

## 3 Tritium isoscapes from 1976 to 2017

The obtained regional gridded amount-weighted annual mean precipitation $^3$H activity time series for the Adriatic–Pannonian region (AP$^3$H_v1 database) capture the well-known decrease in precipitation $^3$H activity (e.g., Fig. 2).

Besides this most striking long-term temporal pattern prevailing in the whole region seen from the isoscapes, subregional differences in precipitation $^3$H activity within the region are also reconstructed. Although no significant relationship was documented between latitude and/or continentality, still-increasing precipitation $^3$H activity was observable inland, with the lowest values documented along the Slovenian and northern Croatian coast in all years (e.g., 1982, 2010, 2014; Fig. 2). This pattern can be related to the generally observed lower activity at maritime coastal stations due to the higher contribution of primary marine evaporation practically free from $^3$H (Eastoe et al., 2012; Rozanski et al., 1991; Tadros et al., 2014; Vreča et al., 2006) and higher contribution of recycled modern meteoric water over the continent. For instance, moisture originating from continental Europe and the Atlantic Ocean was found to be distinct regarding tritium concentrations (8.8 and $\sim 0\,\mathrm{TU}$, respectively; Juhlke et al., 2020).

Two of the longest records from Slovenia (Ljubljana) and Hungary (Budapest) illustrate the performance of the estimations and their potential in mitigating lack of data. Budapest and Ljubljana $^3$H records – complete years were used both in the variograms of the "anchor years" and in the interpolation – were compared to the interpolated product's time series of the nearest grid cell (Fig. 3). In the complete years, when the measured values were used in interpolation, there is an expected perfect match between the measured and modeled values. It becomes clear that the estimated records are more than capable of filling the gaps of the measured time series when there were no measurements (e.g., Ljubljana: 1985 and 1996; Fig. 3b) and usually provide a higher mean annual precipitation $^3$H in the case of incomplete years. The magnitude of differences between the measured and modeled data in the incomplete years varied between $\sim 16.5$ and $1\,\mathrm{TU}$ for Budapest and Ljubljana, with a general tendency of obtaining higher differences with a higher ratio of precipitation not represented by tritium measurements.

The distinctive interannual fluctuation of amount-weighted annual mean $^3$H activity at Budapest and Ljubljana (Fig. 3) also indicates that the AP$^3$H database produced differing subregional variability over the modeled time. In

**Table 1.** Sampling sites with basic geographical information used in the study arranged alphabetically by country code (ISO-3166-1 ALPHA-2) and station name. The number of monthly precipitation $^3$H activity concentration data between 1976 and 2017 used in this study is indicated as well as the number of monthly data available in the Global Network of Isotopes in Precipitation in the same period (column title: GNIP). The stations below the dashed line were used for model performance testing (for details see Sect. 4).

| Country | Station | Latitude (°) | Longitude (°) | Elevation | No. of data (1976–2017) | GNIP |
|---|---|---|---|---|---|---|
| AT | Apetlon | 47.741 | 16.831 | 119 | 321 | |
| AT | Bad Aussee | 47.600 | 13.783 | 640 | 15 | |
| AT | Eisenkappl | 46.489 | 14.584 | 550 | 106 | |
| AT | Gloggnitz | 47.675 | 15.943 | 440 | 96 | |
| AT | Gößl | 47.640 | 13.901 | 710 | 59 | |
| AT | Graz Universität | 47.078 | 15.450 | 366 | 447 | 321 |
| AT | Gutenstein | 47.875 | 15.886 | 475 | 447 | |
| AT | Karlgraben | 47.678 | 15.560 | 775 | 193 | |
| AT | Klagenfurt | 46.643 | 14.320 | 447 | 445 | 318 |
| AT | Lackenhof | 47.870 | 15.142 | 882 | 51 | |
| AT | Nasswald | 47.764 | 15.688 | 774 | 95 | |
| AT | Planneralm | 47.403 | 14.200 | 1605 | 106 | |
| AT | Podersdorf | 47.855 | 16.835 | 120 | 467 | |
| AT | St. Peter im Katschtal | 47.027 | 13.596 | 1220 | 121 | |
| AT | Villacher Alpe | 46.603 | 13.672 | 2164 | 451 | 334 |
| AT | Wien Hohe Warte | 48.249 | 16.356 | 203 | 444 | 421 |
| AT | Wildalpen | 47.664 | 14.978 | 610 | 447 | |
| AT | Zistersdorf | 48.544 | 16.750 | 201 | 88 | |
| HR | Plitvice | 44.881 | 15.619 | 580 | 65 | 27 |
| HR | Puntijarka | 45.908 | 15.968 | 988 | 9 | |
| HR | Tonkovića | 44.790 | 15.368 | 432 | 13 | |
| HR | Zagreb[1] | 45.817 | 15.983 | 157 | 473 | 325 |
| HR | Zavižan | 44.815 | 14.976 | 1594 | 39 | 38 |
| HU | B | 46.070 | 18.111 | 177 | 104 | |
| HU | Boda | 46.087 | 18.047 | 233 | 135 | |
| HU | Budapest | 47.464 | 19.073 | 101 | 287 | 181 |
| HU | Debrecen | 47.475 | 21.494 | 110 | 202 | |
| HU | Het | 46.125 | 18.047 | 165 | 136 | |
| HU | II | 46.100 | 18.093 | 332 | 79 | |
| HU | V | 46.122 | 18.092 | 330 | 60 | |
| HU | Z | 46.037 | 18.125 | 117 | 137 | |
| RS | Belgrade | 44.783 | 20.533 | 243 | 283 | 283 |
| SI | Kozina | 45.604 | 13.932 | 486 | 35 | 35 |
| SI | Kredarica | 46.379 | 13.849 | 2514 | 91 | |
| SI | Ljubljana[2] | 46.095 | 14.597 | 282 | 371 | 291 |
| SI | Murska Sobota | 46.652 | 16.191 | 186 | 11 | |
| SI | Portorož | 45.467 | 13.617 | 2 | 196 | 70 |
| SI | Postojna | 45.766 | 14.198 | 533 | 5 | |
| SI | Rateče | 46.497 | 13.713 | 864 | 88 | |
| SI | Sv. Urban | 46.184 | 15.591 | 283 | 16 | |
| SK | Liptovský Mikuláš | 49.098 | 19.590 | 570 | 96 | 96 |
| HR | Malinska | 45.121 | 14.526 | 1 | 10 | 10 |
| HU | Jósvafő | 48.484 | 20.551 | 211 | 39 | |
| HU | Nick | 47.385 | 17.036 | 139 | 142 | |
| HU | Siófok | 46.911 | 18.041 | 108 | 39 | |
| SI | Zgornja Radovna | 46.428 | 13.943 | 750 | 89 | |

[1] In the investigated period two stations were conducting measurements in Zagreb in a nonoverlapping way (Krajcar Bronić et al., 2020). [2] In the investigated period three stations were conducting measurements in Ljubljana in a nonoverlapping way (Vreča et al., 2014, 2008).

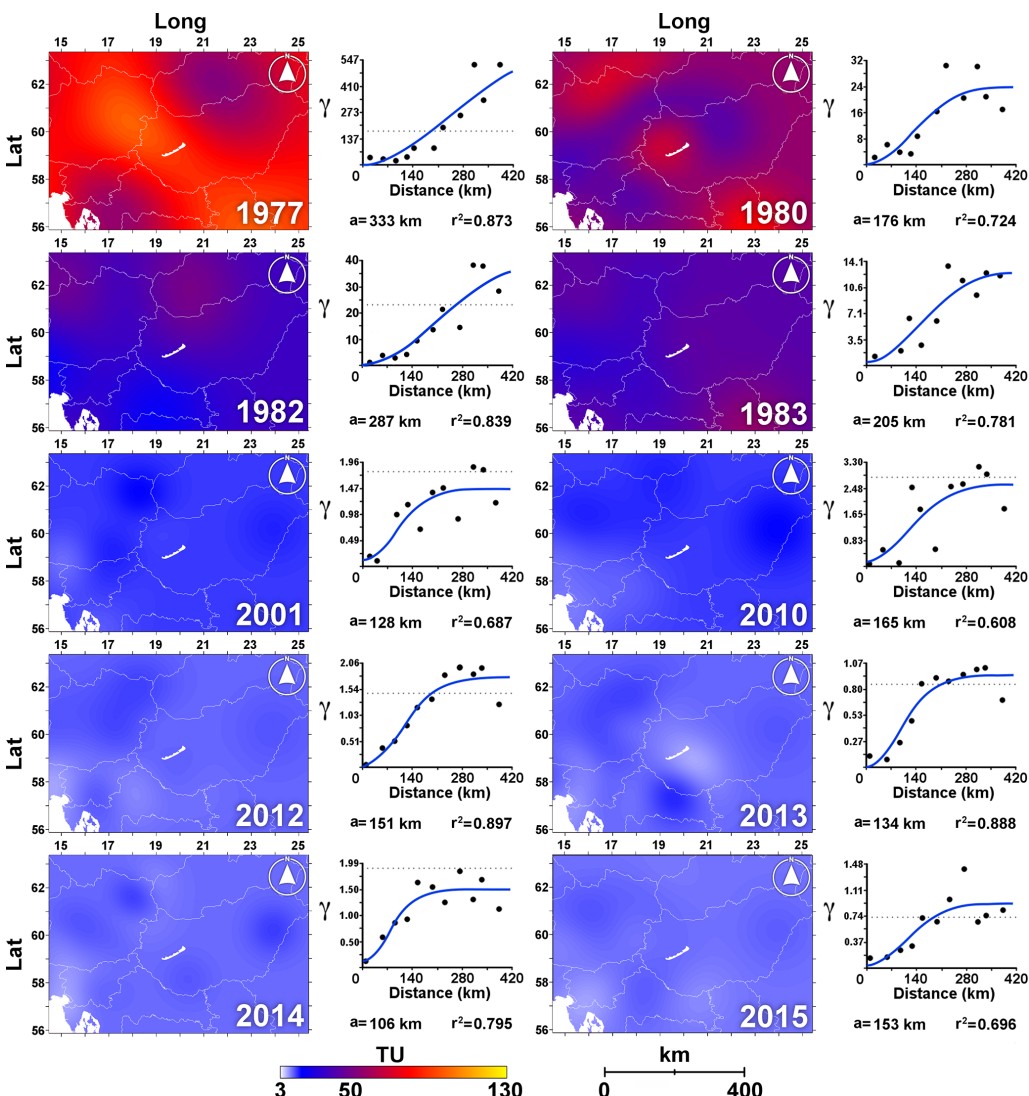

**Figure 2.** Isoscapes of amount-weighted annual mean ³H activity (TU) in precipitation and semivariograms for the reference years in the Adriatic–Pannonian region. The Adriatic Sea and Lake Balaton are marked in white. Isoscape grid resolution: 1 km × 1 km. The figures on the right side of the isoscapes show the empirical (black dots) and theoretical (blue line) semivariograms used for kriging along with the obtained ranges (planar distances in kilometers) and the fit ($r^2$) of the theoretical semivariograms. The dotted horizontal line indicates the average variance.

the whole investigated period (1976–2017) the coefficient of variation in the annual differences between the precipitation ³H values of the complete years in Budapest and Ljubljana was 62 %, calling for the need for spatially representative estimates, such as the ones presented from the AP³H database. In the period when both stations were active at the same time (1981–2003, excluding the incomplete years) the precipitation ³H activity was higher in Budapest than in Ljubljana by 5.5 TU on average, and this difference is quite well reflected by the AP³H data, which were 4.57 TU higher at Budapest on average. Besides these overall differences, distinct interannual variability can be observed at the Ljubljana and Budapest stations. For example, the maxima of the

modeled precipitation ³H activity occurred in 1976 at the Ljubljana station and in 1978 at the Budapest station, and a minor peak seen in 1988 at Budapest (Fig. 3a) was without a counterpart in the Ljubljana records (Fig. 3b).

The estimated effective range shows a decrease in the spatial autocorrelation of tritium activity concentration of precipitation from the 1970s to the 2010s: ∼ 440 km in the 1970s, ∼ 425 km in the 1980s, and ∼ 360 and 235 km in the 2000s and 2010s, respectively (Fig. 4). This period (1970–2010) was characterized by the removal of bomb tritium from the atmosphere (Araguas-Araguas et al., 1996; Palcsu et al., 2018). The overwhelming activity of bomb-produced ³H was several orders of magnitude higher than the natural

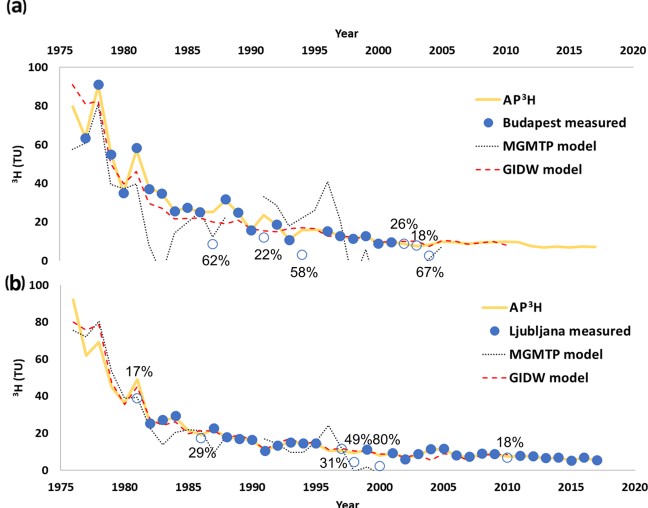

**Figure 3.** Measured and estimated $^3$H values at **(a)** Budapest, Hungary, and **(b)** Ljubljana, Slovenia, between 1976 and 2017. The dotted black lines indicate the estimations of the "modified global model of tritium in precipitation" (MGMTP; Zhang et al., 2011), and the dashed red ones indicate the global inverse distance weighted model (GIDW; Jasechko and Taylor, 2015). Note here that uninterpretable negative estimates of the MGMTP were not shown. The empty circles indicate an incomplete year, in which the given $^3$H value was not used for interpolation in the AP$^3$H database, and the percentages next to these symbols indicate the ratio of fallen precipitation not analyzed for $^3$H in a given year.

background (Rozanski et al., 1991) and largely masked the smaller-scale natural variability. During recent decades the tritium activity in precipitation has declined globally and regionally, approaching the natural prebomb levels, indicating that bomb tritium is now barely present in modern precipitation. Since in the Adriatic–Pannonian region $^3$H activity in precipitation approached natural levels by the 2000s (Krajcar Bronić et al., 2020; Palcsu et al., 2018; Vreča et al., 2008), it can be expected that the $\sim$ 200–300 km range obtained for the 2010s reflects the range of similarity of natural $^3$H variability in the study area (southeastern Europe and eastern Central Europe).

## 4  Verification of goodness of interpolation

The performance of the AP$^3$H database replicating measured precipitation $^3$H activity was tested via out-of-sample verification. The actual amount-weighted annual mean precipitation $^3$H activity time series at a Hungarian (Nick, Fig. 5a) and a Slovenian (Zgornja Radovna, Fig. 5b) station were compared to the AP$^3$H data of the grid cell closest to the specific stations. The annual mean precipitation $^3$H activity estimates from the AP$^3$H database fit the amount-weighted annual mean precipitation $^3$H activity data nicely for both independent records even if a small portion ($< 8\%$) was not rep-

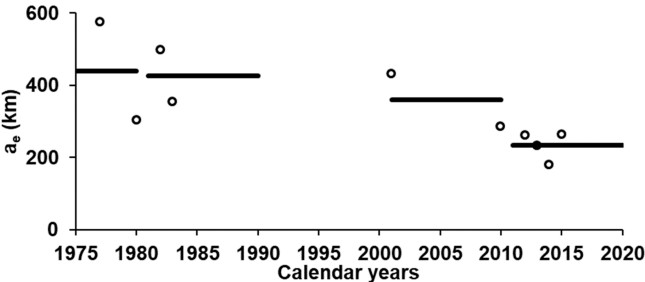

**Figure 4.** Effective ranges ($a_e$) of the semivariograms used for kriging. Annual ranges are indicated by empty circles, and the decadal averages are indicated by horizontal black lines.

resented with measured $^3$H in the experimental data (Fig. 5a, b). The agreement between the independent mean annual precipitation $^3$H record and the AP$^3$H estimates is superb for Zgornja Radovna, where the difference between modeled and actual values ranged from 1 % to 6 % (mean: 3.2 %). The difference between modeled and actual values in complete years for Nick ranged from 0 % to 31 % (mean: 8.3 %). The relatively larger mean error for the Nick record is due to an overestimation of the model values in 1994 (Fig. 5a); however, the model data fit the rest of the measured data as well as is seen for Zgornja Radovna. The pattern observed for the incomplete years strengthens previous impressions that (i) modeled mean annual precipitation $^3$H is higher than the mean calculated from just a few monthly measurements, and (ii) there is a tendency of obtaining higher differences with a higher ratio of precipitation not represented by tritium measurements. The comparison between the modeled and actual values for the stations' recorded precipitation $^3$H for multiple but exclusively incomplete years further strengthens these impressions.

Station records from Nick and Zgornja Radovna were not used in the interpolation to derive the AP$^3$H database, yet these perfectly match the actual values when $^3$H activity concentration was measured for the total volume of annual precipitation. This excellent agreement supports the assumption that the AP$^3$H database provides similarly accurate estimates as at other stations for the incomplete years as well. If this is the case, then differences between the AP$^3$H data and the mean precipitation $^3$H activity estimates accompanied by a certain portion of precipitation amount lacking $^3$H activity can represent a weighting bias due to the fragmented observation record. Comparing the differences between (i) the estimated mean precipitation $^3$H activity values from these fragmented records and the AP$^3$H estimates with (ii) the percentage of precipitation amount lacking $^3$H activity data in the incomplete years for Zgornja Radovna, Nick, Jósvafő, and Malinska revealed the relationship between these (Fig. 5e). If $^3$H activity was lacking for less than 15 % of the annual precipitation total, the difference between the calculated mean precipitation $^3$H activity values and the AP$^3$H estimates is prac-

tically negligible (Fig. 5e). The difference between modeled and calculated mean precipitation $^3$H considerably increased (> 6 TU) if the proportion of the precipitation lacking $^3$H activity was between 15 % and 30 % of the annual precipitation total (Fig. 5e). According to the GNIP protocol the required completeness is 70 % of the precipitation total for calculating amount-weighted mean isotopic values for a certain period (IAEA, 1992). The above results (Fig. 5e) confirm that a proportion of missing data of 30 % is indeed a critical limit because, if exceeded, each calculated mean precipitation $^3$H activity value of these scanty annual datasets showed great deviation from the modeled values. However, occasionally a precipitation deficit over 15 % yet under 30 % causes a noticeable difference (Fig. 5e), suggesting that a proportion of missing data of less than 30 % may be advisable for the calculation of representative amount-weighted isotopic means. In addition, these reinforce the decision made in the present study to only include values for which more than 85 % of fallen precipitation was analyzed for $^3$H.

Although at Nick and Zgornja Radovna – as seen above – the AP$^3$H database provides a realistic estimation (Fig. 5a, b), one may ask why the calculated annual mean precipitation $^3$H activity values are systematically higher at Siófok (mean difference in the complete years: 1.52 TU) than the AP$^3$H data (Fig. 5d). It could be explained by the closeness of the largest shallow lake in Central Europe, Lake Balaton (Hatvani et al., 2020). The mean residence time in the largest basin of the lake, Siófok basin, was estimated to be between 2 and 6 years in the 1990s (Istvánovics et al., 2002), which is presumably in the same range in the 2010s as well. Keeping in mind the gradual decrease in $^3$H in meteoric waters (in the region; e.g., Fig. 4), the local evaporation from this "aged" reservoir can provide an isotopically detectable contribution to the atmospheric moisture, which can be transferred to the falling rain droplets via molecular exchange mechanisms (Bolin, 1959), resulting in higher tritium activity values at the Siófok station than the ones from the AP$^3$H database (Fig. 5d). This is a simple example illustrating the potential of this new regional annual mean precipitation $^3$H activity database (AP$^3$H_v1) in hydrological applications.

## 5   Comparison between global and regional modeled annual mean $^3$H activity concentrations in precipitation across the Adriatic–Pannonian region

The presented AP$^3$H_v1 database of tritium activity was compared with the spatially corresponding output of both currently available global precipitation tritium isoscapes: the modified global model of tritium in precipitation (MGMTP; Zhang et al., 2011) and the global inverse distance weighted model (GIDW) at Budapest (Fig. 3a) and Ljubljana (Fig. 3b). Between 1975 and 1980 the estimates from the AP$^3$H database and the MGMTP are very similar and resemble

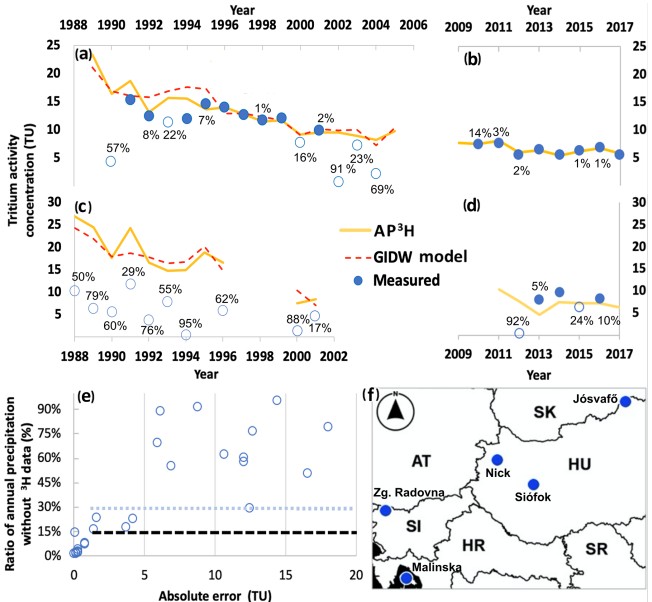

**Figure 5.** Amount-weighted annual mean $^3$H activity concentration in precipitation measured at **(a)** Nick (1990–2004; HU), **(b)** Zgornja Radovna (2010–2017; SI), **(c)** Jósvafő (HU; 1988–1996) and Malinska (2000–2001; Krk Island; HR), and **(d)** Siófok (2009–2017; HU) stations along with the corresponding AP$^3$H estimates. The percentages next to the symbols of the measured values indicate the ratio of fallen precipitation not analyzed for $^3$H in a given year if it was > 0 %. The dashed red line in panels **(a)** and **(c)** shows the corresponding values of global inverse distance weighted model (GIDW; Jasechko and Taylor, 2015). **(e)** Cross plot showing the relationship between absolute error of estimated precipitation $^3$H activity by the AP$^3$H and the percentage of precipitation without $^3$H measurement at Nick, Zgornja Radovna, Jósvafő, and Malinska. The thick dashed line marks the limit of completeness (15 %) applied in this study, while the gray dotted line marks the limit of completeness (30 %) according to the GNIP protocol (IAEA, 1992). **(f)** The map shows the location of the five sites used for evaluation of the performance of the AP$^3$H database.

the actual weighted annual mean precipitation $^3$H at Budapest. However, only at Ljubljana is the MGMTP capable of steadily reproducing the actual measurements until the late 1990s. Afterwards, it indicates solely negative values, which are uninterpretable, just as most of the MGMTP-predicted values at Budapest after 1980. Meanwhile, the AP$^3$H database provided much more accurate and reliable results (Fig. 3) as discussed above. Note here that the weak estimation of the MGMTP can be attributed to the difficulties in reading the precipitation tritium activity values from the only available output (isoline map) of the model and the undocumented factors of the model in given years.

The GIDW (Jasechko and Taylor, 2015) model was capable of reproducing the measured precipitation tritium values much more accurately at all locations than the MGMTP (Fig. 3). Nevertheless, the GIDW model produced a strik-

ing overestimation at the beginning of the modeled period, for example in 1977, when the measured values at Budapest were overestimated by > 20 TU TS1 (Fig. 3a). In contrast, the GIDW model underestimated the actual values from 1981 to 1991 except for the year 1 year TS2 (Fig. 3a).

The skills of the AP$^3$H database and the GIDW model could only be compared at the Nick station because the record of Zgornja Radovna starts in the terminal year of the GIDW model's coverage. In the first complete year of the Nick record the GIDW estimation is closer to the actual measurements compared to the AP$^3$H estimations; however, from 1992 to 1994 the GIDW estimations show much higher (mean error: 4.6 TU) annual mean $^3$H activity compared to the values calculated from the monthly measurements. Meanwhile, the difference from the AP$^3$H estimations is smaller (mean error: 2.2 TU). Interannual variations of the estimated precipitation $^3$H activity from the GIDW and the AP$^3$H database are in agreement at Jósvafő, and the models are largely in agreement regarding the absolute values and the decreasing trend, too.

However, the MGMTP had practically no success in estimating the actual measured values. For instance, in both 2000 and 2001 the MGMTP produced negative – thus meaningless – values at Malinska, when direct measurements were available, while the present database provided more reliable estimates (Fig. 5c).

Taken all together, the AP$^3$H_v1 database gave better regional estimates than either of the global models. This may be because (i) the AP$^3$H database is based on more local station data for the study area compared to the global models relying only on the GNIP records (Table 1), and (ii) the improved performance of the database obviously benefitted from incorporating the spatial correlation structure of precipitation tritium activity into the presented geostatistical prediction.

## 6   Data availability

The final product (AP3H_v1.nc), the spatially continuous annual (1976–2017) 1 km × 1 km grids of amount-weighted annual mean precipitation tritium activity for the Adriatic–Pannonian region, is provided in a netCDF-4 (Network Common Data Form) format available at PANGAEA (https://doi.org/10.1594/PANGAEA.896938; Kern et al., 2019) and compiled using the EPSG 3857 projection. For a solely visual inspection of the annual grids Panoply (NASA, 2020) is recommended. An R script written to be able to browse the dataset and convert the projection to EPGS 4326 is provided in the Supplement. This publication describes the AP$^3$H_v1 database. As the database is updated with corrected or extended new data the new versions will be indexed with incremental integers.

## 7   Possibility of applications (outlook and conclusions)

Continuous long-term records of tritium in precipitation are scarcely available worldwide; thus estimation or modeling is necessary to exploit its potential in hydrological research. In order to decrease the uncertainty of tritium activity in the hydrological models, the application of regional $^3$H models has to be increased since these are more capable of producing accurate estimations than global ones (Stewart and Morgenstern, 2016).

Instead of using remote station data or ad hoc composite curves, site-specific time series retrieved from the presented AP$^3$H_v1 database of amount-weighted annual mean precipitation $^3$H isoscapes should be used. These isoscapes (Kern et al., 2019) can serve as a reference dataset for studies on infiltration dynamics, water transport through various compartments of the hydrological cycle, mixing processes, and run-off modeling, for example, to estimate mean residence time in surface waters and groundwater (Kanduč et al., 2014; Ozyurt et al., 2014; Szucs et al., 2015). As a specific type of hydrogeological application, the regional model of $^3$H time series will serve as a benchmark in estimating the mean infiltration age of drip water (Kluge et al., 2010), which can provide an additional tool for ongoing cave monitoring studies from the region (e.g Czuppon et al., 2018, 2013; Fehér et al., 2016; Surić et al., 2010) in a spatiotemporally accurate way.

The higher precipitation $^3$H activity observed at a lakeshore station (Fig. 5c) likely reflects moisture recycling from the aged lake surface water via evaporation to local precipitation. The observed deviation highlights the potential of the database to reveal subregional anomalous local sources in the hydrological cycle. As a special case the post-2010 isoscapes can serve as benchmarks for background tritium activity for the region, helping to determine local increases in technogenic tritium from these backgrounds.

The AP$^3$H database was able to provide better estimates than either of the currently available global models for the study area. Its values seem more capable of reproducing the actual annual mean precipitation $^3$H activity than the fragmented $^3$H data, which represent less than 85 % of the total annual precipitation. Prior to 1975 we encourage the use of the GIDW model's estimations (Jasechko and Taylor, 2015) as a reference for studies dealing with precipitation tritium activity. The AP$^3$H and the GIDW model data should be spliced together in 1975 and can be used together in the need of a semicentennial precipitation tritium activity dataset.

**Video supplement.** Isoscape of precipitation amount-weighted annual mean tritium ($^3$H) activity from 1976 to 2017 for the Adriatic–Pannonian region. https://av.tib.eu/media/47154 TS3

**Supplement.** The supplement related to this article is available online at: https://doi.org/10.5194/essd-12-1-2020-supplement.

**Author contributions.** ZK designed the experiments. PV, MŠ, TK, LP, MS, GC, and IKB contributed data. IGH and DE developed the model code and performed the analyses. ZK, IGH, PV, and DE prepared the manuscript with contributions from TK, MŠ, IF, and BK. The authors applied the FLAE approach for the sequence of authors. All authors took part in the manuscript preparation and revision.

**Competing interests.** The authors declare that they have no conflict of interest.

**Acknowledgements.** This work was supported by the National Research, Development and Innovation Office under grants SNN118205 and PD121387 and by the Slovenian Research Agency ARRS under grants N1-0054, J4-8216, and P1-0143. The research was partly supported by the European Union and the State of Hungary and cofinanced by the European Regional Development Fund in the GINOP-2.3.2-15-2016-00009 project "ICER". The authors thank all the people who participated in precipitation sampling and measurements. This is contribution no. 66 of the 2ka Paleoclimate Research Group.

**Financial support.** This research has been supported by the National Research, Development and Innovation Office (grant nos. SNN118205 and PD121387), the Slovenian Research Agency ARRS (grant nos. N1-0054, J4-8216, and P1-0143), the European Union, and the State of Hungary and was cofinanced by the European Regional Development Fund (grant no. GINOP-2.3.2-15-2016-00009 "ICER").

**Review statement.** This paper was edited by Jens Klump and reviewed by two anonymous referees.

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

## Remarks from the language copy-editor

CE1 Fewer is correct here as "years" is a countable noun.

## Remarks from the typesetter

TS1 Please give an explanation of why this needs to be changed. We have to ask the handling editor for approval. Thanks.

TS2 Please give an explanation of why this needs to be changed. We have to ask the handling editor for approval. Thanks.

TS3 Please note that it cannot be used as an abstract as an abstract should consist of a video where the authors summarize their paper, which is not the case here. Therefore please provide a short text in full sentences describing what can be seen in the video, being sure to mention the DOI and the reference for the DOI here. The current information inserted here does not meet the requirements.