# Peer review of "Isoscape of precipitation amount-weighted annual mean tritium $(^{3}H)$ activity from 1976 to 2017 for the Adriatic-Pannonian region - $AP^{3}H_{v}1$ database"

_Earth System Science Data, 2019_

## Referee Comment (RC1) · Anonymous Referee #1 · 15 Mar 2020

The manuscript describes a high-resolution gridded dataset for tritium in precipitation across the Adriatic-Pannonia region in Europe. I am not familiar with the applications of tritium for hydrology, my expertise is on geostatistical methods for hydrological sciences.

The objective of the work is clearly stated in the abstract and in the introduction. Material and methods are described in detail in section 2. The data sources used are properly reported. Standard statistical techniques, such as ordinary kriging, have been used for spatial analysis. Pros and cons of the applied statistical methods are discussed in detail, especially in connection with the scarcity of data available. Section 3

described the gridded dataset. Section 4 contains the evaluation: the regional dataset has been compared with global ones, the benefits are clearly highlighted; a validation against independent measurements is also included (see Fig.5). All the presented results support the conclusions of the authors presented in section 5.

The contribution of this study for regional hydrological applications is valuable, since the uniqueness of such a reference and up-to-date dataset. Given the limited amount of stations available, the creation of a gridded dataset is totally justified and can provide useful data where no direct measurements are available. The statistical analysis is, as far as I can judge, without major flaws. The presentation of the manuscript is clear and concise.

In conclusion, the study is valuable. My advice to the editor is to publish the manuscript after minor adjustments to the text. Specific comments follow.

Comments:

- Why use such a high-resolution 1x1 km grid when the planar distances (Fig.3) are hundreds of kilometers? By using this grid, the authors implicitly persuade the users that the information is available on a very local scale. This is not the case. The authors need to (1) justify their choice of a 1x1 km grid; (2) explicitly state that their gridded dataset is suitable for the representation of variations in the field over much larger spatial scales than the grid spacing.

- The authors apply kriging without showing that the input data satisfies the prerequisites for a direct application of ordinary kriging. However, the validation shows that the output is useful and -in a sense- this justifies the application of kriging. My question for you is: have you considered other statistical interpolation methods? What is the reason that made you choose kriging?

- Figure 3. This is perhaps the core result of the paper and I like very much the way the authors present it. However, the blue shades in the colour scale are by

far not optimal in representing the fields. Please present your main results in a way that the readers can fully appreciate them.

---

## Referee Comment (RC2) · Anonymous Referee #2 · 26 Apr 2020

The authors have found a good way to fill in gaps in the existing information by creating a statistical model which uses the 3H data collected from Austria and the northern Balkans. The manuscript presents a time-based 3H precipitation isoscape of North-Balkan. The text is well-structured and easy to read. However, I do have some comments.

Key words: why do the authors only mention Slovenia and Hungary and not the northern Balkans as the title suggests?

The purpose of this work is not only to create a database, but also to analyse and draw conclusions. I would recommend to slightly expand the purpose in the introduction.

The authors need to explain why they have used the 1×1 km grids. This seems an unreasonable accuracy compared to the size of the study area.

"..the largest shallow freshwater lake in Central Europe". Wouldn't the 'largest lake in Central Europe' already be enough?

It is hard to follow the isoscape in Figure 3: I would recommend to use red-blue instead of the current green-blue combination to improve the contrast.

The used data base (https://doi.pangaea.de/10.1594/PANGAEA.896938) is presented in a less used format (my computer required additional software to read it). Wouldn't it be possible to present it in the HTML format to make it more usable? An example: https://doi.pangaea.de/10.1594/PANGAEA.911474?format=html#download.

There seems to be some confusion with the parentheses. I've highlighted these in the attached file.

Please also note the supplement to this comment:
https://www.earth-syst-sci-data-discuss.net/essd-2019-244/essd-2019-244-RC2-supplement.pdf
* * *
[Figure]

**Supplement:**

[revised manuscript text omitted]

---

## Author Comment (AC2) · 14 May 2020

Dear Reviewer,

Please find your answers to your questions below.

Yours sincerely,

The authors

https://www.earth-syst-sci-data-discuss.net/essd-2019-244/#discussion

Anonymous Referee #2 The authors have found a good way to fill in gaps in the existing

information by creating a statistical model which uses the 3H data collected from Austria and the northern Balkans. The manuscript presents a time-based 3H precipitation isoscape of North-Balkan. The text is well-structured and easy to read. However, I do have some comments. We would like to thank Reviewer#2 for her/his positive opinion on our work and for the constructive comments.

————————— Comment-1: Key words: why do the authors only mention Slovenia and Hungary and not the northern Balkans as the title suggests?

Response-1: Our intention was to not repeat the words in the title among the key words, but mentioning the countries covered entirely by the developed database.

————————— C-2: The purpose of this work is not only to create a database, but also to analyse and draw conclusions. I would recommend to slightly expand the purpose in the introduction.

R-2: The introduction has been slightly and the discussion thoroughly extended. In addition, the paper was partially restructured during the revision.

————————— C-3: The authors need to explain why they have used the 1×1 km grids. This seems an unreasonable accuracy compared to the size of the study area.

R-3: The 1×1 km grid resolution was chosen based on practical considerations, it does not aim to imply that there are such fine km-scale differences, yet help the users to delineate smaller outcrops (e.g. watersheds) more accurately. This explanation will be added to the revised manuscript.

————————— C-4: "..the largest shallow freshwater lake in Central Europe". Wouldn't the 'largest lake in Central Europe' already be enough?

R-4: We accept that "freshwater" can be omitted however we would like to keep "shallow". We think this information is useful because the 2 to 6 yrs residence time for the water mentioned in the next sentence is understandable if the lake is shallow however might be weird if someone imagine a deep lake based on the shortened term "largest

lake in Central Europe".

——————— C-5: It is hard to follow the isoscape in Figure 3: I would recommend to use red-blue instead of the current green-blue combination to improve the contrast.

R-5: Accepted. This figure has been revised substantially. We hope the applied more complex color scale (white-blue-red-yellow) does improve the contrast in the map series sufficiently.

——————— C-6: The used data base (https://doi.pangaea.de/10.1594/PANGAEA.896938) is presented in a less used format (my computer required additional software to read it). Wouldn't it be possible to present it in the HTML format to make it more usable? An example: https://doi.pangaea.de/10.1594/PANGAEA.911474?format=html#download .

R-6: The database can be considered as temporal sequence of 3D data quite similar to meteorological data. The chosen netCDF format is a common and popular file format in meteorology. The suggested example (https://doi.pangaea.de/10.1594/PANGAEA.911474 ) is a time series from a single station so, we think, it is not an applicable analogue to the presented dataset. To facilitate the usage of the data set an R-script was already provided in the supplement. In addition, we will expand the "Data format and availability" section with a sentence in which we will provide a link to the freeware tool of NASA (Panoply: https://www.giss.nasa.gov/tools/panoply/) with which interested readers can visualize and inspect of the netCDF files of annual grids.

——————— C-7: There seems to be some confusion with the parentheses. I've highlighted these in the attached file.

R-7: We have carefully formatted the citations in the main text to correct the superfluous parentheses. The missing spaces between values and dimensions are also corrected at each highlighted place.

---

## Author Response (AR2)

**Research Centre for Astronomy and Earth Sciences**
**Institute for Geological and Geochemical Research**

H-1112 Budapest, Budaörsi út 45.   Tel.: (+36) 1 319-3137   www.csfk.mta.hu

[Figure]

Date: 19.05.2020
Subject: **Submission of revised MS**

Dear Prof. Klump,

Please find enclosed our revised manuscript now entitled: '*Isoscape of precipitation amount-weighted annual mean tritium ($^3H$) activity from 1976 to 2017 for the Adriatic-Pannonian region - AP$^3$H_v1 database*.

Please note that during the revision an additional colleague has contributed to the study (Miklós Süveges) and has been added as an author to the title page and the 'Author contribution' section as well. In adiition, considering the contribution to the revision the author-order has been changed as well.

On behalf of the team of authors I state here that the material in the manuscript is new work none of the submitted material has been published or is under consideration elsewhere, including the Internet, and will not be submitted to any other journal while the present manuscript is under consideration at ESSD.

If you have any questions regarding the enclosed materials, please do not hesitate to contact me.

Thank you for your editorial guidance.

Yours sincerely,

István Gábor Hatvani

https://www.earth-syst-sci-data-discuss.net/essd-2019-244/#discussion

**Anonymous Referee #1**

**The manuscript describes a high-resolution gridded dataset for tritium in precipitation across the Adriatic-Pannonia region in Europe. I am not familiar with the applications of tritium for hydrology, my expertise is on geostatistical methods for hydrological sciences.**
**The objective of the work is clearly stated in the abstract and in the introduction. Material and methods are described in detail in section 2. The data sources used are properly reported. Standard statistical techniques, such as ordinary kriging, have been used for spatial analysis. Pros and cons of the applied statistical methods are discussed in detail, especially in connection with the scarcity of data available. Section 3 described the gridded dataset. Section 4 contains the evaluation: the regional dataset has been compared with global ones, the benefits are clearly highlighted; a validation against independent measurements is also included (see Fig.5). All the presented results support the conclusions of the authors presented in section 5.**
**The contribution of this study for regional hydrological applications is valuable, since the uniqueness of such a reference and up-to-date dataset. Given the limited amount of stations available, the creation of a gridded dataset is totally justified and can provide useful data where no direct measurements are available. The statistical analysis is, as far as I can judge, without major flaws. The presentation of the manuscript is clear and concise.**
**In conclusion, the study is valuable. My advice to the editor is to publish the manuscript after minor adjustments to the text.**
We would like to thank Reviewer#1 for her/his positive opinion on our work and for the constructive comments.

**Specific comments follow.**
**Comment-1: Why use such a high-resolution 1x1 km grid when the planar distances (Fig.3) are hundreds of kilometers? By using this grid, the authors implicitly persuade the users that the information is available on a very local scale. This is not the case. The authors need to (1) justify their choice of a 1x1 km grid; (2) explicitly state that their gridded dataset is suitable for the representation of variations in the field over much larger spatial scales than the grid spacing.**
Response-1: Thank you for the suggestion. The 1×1 km grid resolution was chosen based on practical considerations, it does not aim to imply that there are such fine km-scale differences, yet help the users to delineate smaller outcrops (e.g. watersheds) more accurately. This explanation has been added to the MS, see the track changes version in lines 220-225.

**C-2: The authors apply kriging without showing that the input data satisfies the prerequisites for a direct application of ordinary kriging. However, the validation shows that the output is useful and -in a sense- this justifies the application of kriging. My**

**question for you is: have you considered other statistical interpolation methods? What is the reason that made you choose kriging?**

R-2: To further reinforce the Reviewer's opinion on that the verification employed in the study is convincing two additional stations have been included as out-of-sample verification (please see Fig. 5). In addition, the discussion has been extended and the MS reorganized.

An additional checking was performed on the amount weighted annual means using h-scattergrams (Bohling, 2005) which did not find any outliers that have been introduced by the weighting procedure confirming that the dataset satisfies certain prerequisites of kriging. This explanation has been added to the track changes version of the revised manuscript in lines 160-164.

The deviance from normal distribution in the case of the $^3H$ values was found negligible; thus kriging can be applied confidently to the data. Moreover, despite kriging is known to smooth the data, thus decrease the range of the actual values, in the present case the extremities (positive or negative) were not the main subject of the analysis, rather the mid 60% of the data. This was one of the main reasons for choosing kriging to investigate the large-scale patterns, which kriging is highly applicable for (e.g. Wackernagel, 2003 or Chilès and Delfiner, 2012).

**C-3: Figure 3. This is perhaps the core result of the paper and I like very much the way the authors present it. However, the blue shades in the colour scale are by far not optimal in representing the fields. Please present your main results in a way that the readers can fully appreciate them.**

R-3: Fig. 3 has been substantially changed in the revised version. We hope that the more complex color scale (white-blue-red-yellow) sufficiently improves the contrast in the map series. See Fig. 2 in line 250.

https://www.earth-syst-sci-data-discuss.net/essd-2019-244/#discussion

**Anonymous Referee #2**
**The authors have found a good way to fill in gaps in the existing information by creating a statistical model which uses the 3H data collected from Austria and the northern Balkans.  The manuscript presents a time-based 3H precipitation isoscape of North-Balkan.  The text is well-structured and easy to read.  However, I do have some comments.**

We would like to thank Reviewer#2 for her/his positive opinion on our work and for the constructive comments.

**Comment-1: Key words: why do the authors only mention Slovenia and Hungary and not the northern Balkans as the title suggests?**

Response-1: Our intention was to not repeat the words in the title among the key words, but mentioning the countries covered entirely by the developed database.

**C-2: The purpose of this work is not only to create a database, but also to analyse and draw conclusions. I would recommend to slightly expand the purpose in the introduction.**

R-2: The introduction (lines 82-85) and the discussion have been extended and latter restructured during the revision, please see track changes version.

**C-3: The authors need to explain why they have used the 1×1 km grids.  This seems an unreasonable accuracy compared to the size of the study area.**

R-3: The 1×1 km grid resolution was chosen based on practical considerations, it does not aim to imply that there are such fine km-scale differences, yet help the users to delineate smaller outcrops (e.g. watersheds) more accurately. This explanation has been added to the MS, see the track changes version in lines 220-225.

**C-4: "..the largest shallow freshwater lake in Central Europe".  Wouldn't the 'largest lake in Central Europe' already be enough?**

R-4: We accept that "freshwater" can be omitted however we would like to keep "shallow". We think this information is useful because the 2 to 6 yrs residence time for the water mentioned in the next sentence is understandable if the lake is shallow however might be weird if someone image a deep lake based on the shortened term "largest lake in Central Europe".

**C-5: It is hard to follow the isoscape in Figure 3: I would recommend to use red-blue instead of the current green-blue combination to improve the contrast.**

R-5: Accepted. This figure has been revised substantially. We hope the applied more complex color scale (white-blue-red-yellow) does improve the contrast in the map series sufficiently. See Fig. 2 in line 250.

**C-6: The used data base (https://doi.pangaea.de/10.1594/PANGAEA.896938) is presented in a less used format (my computer required additional software to read it).  Wouldn't it**

**be possible to present it in the HTML format to make it more usable?  An example: https://doi.pangaea.de/10.1594/PANGAEA.911474?format=html#download .**

R-6: The database can be considered as temporal sequence of 3D data quite similar to meteorological data. The chosen netCDF format is a common and popular file format in meteorology. The suggested example (https://doi.pangaea.de/10.1594/PANGAEA.911474 ) is a time series from a single station so, we think, it is not an applicable analogue to the presented dataset. To facilitate the usage of the data set an R-script was already provided in the supplement. In addition, we have expanded the "Data format and availability" section with a sentence in which we provide a link to the freeware tool of NASA (Panoply: https://www.giss.nasa.gov/tools/panoply/) with which interested readers can visualize and inspect of the netCDF files of annual grids (line 483)

**C-7: There seems to be some confusion with the parentheses. I've highlighted these in the attached file.**

R-7: We have carefully formatted the citations in the main text to correct the superfluous parentheses. The missing spaces between values and dimensions are also corrected at each highlighted place.

[revised manuscript text omitted]

Törölt cellák
Beszúrt cellák
Beszúrt cellák
Beszúrt cellák

Törölt cellák
Beszúrt cellák

| | | | | | | | |
|---|---|---|---|---|---|---|---|
| AT | Planneralm | 47.403 | 14.200 | 1605 | AT | 106 | |
| AT | Podersdorf | 47.855 | 16.835 | 120 | AT | 467 | |
| AT | St. Peter im Katschtal | 47.027 | 13.596 | 1220 | AT | 121 | |
| AT | Villacher Alpe | 46.603 | 13.672 | 2164 | AT | 451 | 334 |
| AT | Wien Hohe Warte | 48.249 | 16.356 | 203 | AT | 444 | 421 |
| AT | Wildalpen | 47.664 | 14.978 | 610 | AT | 447 | |
| AT | Zistersdorf | 48.544 | 16.750 | 201 | AT | 88 | |
| HR | Plitvice | 44.881 | 15.619 | 580 | HR | 65 | 27 |
| HR | Puntijarka | 45.908 | 15.968 | 988 | | 9 | |
| HR | Tonkovića | 44.790 | 15.368 | 432 | | 13 | |
| HR | Zagreb[1] | 45.817 | 15.983 | 157 | HR473 | 133325 | |
| HR | Zavižan | 44.815 | 14.976 | 1594 | HR | 39 | 38 |
| HU | B | 46.070 | 18.111 | 177 | | 104 | |
| HU | Boda | 46.087 | 18.047 | 233 | | 135 | |
| HU | Budapest | 47.464 | 19.073 | 101 | HU287 | 181 | |
| HU | Debrecen | 47.475 | 21.494 | 110 | HU | 202 | |
| | Met B | 46.070 | 18.111 | 177 | HU | 41 | |
| | Met Boda | 46.087 | 18.047 | 233 | HU | 82 | |
| HU | Met Het | 46.125 | 18.047 | 165 | HU136 | 82 | |
| HU | Met II. üz | 46.100 | 18.093 | 332 | HU79 | 27 | |
| HU | Met V. üz | 46.122 | 18.092 | 330 | HU60 | 58 | |
| HU | Met Z | 46.037 | 18.125 | 117 | HU137 | 75 | |
| RS | Belgrade | 44.783 | 20.533 | 243 | RS283 | 283 | |
| SI | Kozina | 45.604 | 13.932 | 486 | SI35 | 35 | |
| SI | Kredarica | 46.379 | 13.849 | 2514 | SI | 91 | |
| SI | Ljubljana[2] | 46.095 | 14.597 | 282 | SI | 371 | 291 |
| SI | Murska Sobota | 46.652 | 16.191 | 186 | SI | 11 | |
| SI | Portorož | 45.467 | 13.617 | 2 | SI | 196 | 70 |
| SI | Postojna | 45.766 | 14.198 | 533 | SI | 5 | |
| SI | Rateče | 46.497 | 13.713 | 864 | SI | 88 | |
| SI | Sv. Urban | 46.184 | 15.591 | 283 | SI | 16 | |
| SK | Liptovský Mikuláš | 49.098 | 19.590 | 570 | SK96 | 96 | |

Marginal annotations: Beszúrt cellák / Törölt cellák / Beszúrt cellák / Beszúrt cellák / Törölt cellák / Beszúrt cellák / Beszúrt cellák / Törölt cellák / Beszúrt cellák

[revised manuscript text omitted]